# Development of Effective Infrared Reflective Coatings

Józsefné Mara [1], Attila-Ede Bodnár [1], László Trif [2] and Judit Telegdi [1,2,3,*]

1. EGROKORR Ltd., Fehérvári út 63–65, 2030 Érd, Hungary; maram@egrokorr.hu (J.M.);
   bodnara@egrokorr.hu (A.-E.B.)
2. Research Centre for Natural Sciences, Magyar Tudósok Körútja 2, 1117 Budapest, Hungary; trif.laszlo@ttk.hu
3. Faculty of Light Industry and Environmental Engineering, Óbuda University, Doberdó u. 6,
   1034 Budapest, Hungary
* Correspondence: telegdi.judit@ttk.hu

**Featured Application: This infrared reflective coating could be applied to vehicles and buildings in order to save energy.**

**Abstract:** The adsorption of surfaces exposed to sunlight results in increased temperatures that can cause physical damage and an increase in energy consumption. Infrared reflective coatings can keep objects cooler and have significant benefits in a wide variety of application by reflecting infrared light and decreasing heat, reducing operating costs, improving energy efficiency in buildings and vehicles, and extending an objects' lifespan. The main aim of our research was to develop coatings in a RAL7016 Anthracite grey color with minimum heat adsorption in the infrared wavelength range. This was achieved using a combination of infrared transparent and infrared reflective pigment built-in coatings applied on two primers: white and black. Infrared reflectivity or transparency, as well as surface temperature, was investigated as a function of the composition and concentration of pigments. These coatings were characterized by chromatic parameters, by total solar and infrared solar reflectance in the UV, visible, and infrared wavelength range, and by heat reflection. Among the coatings developed, two produced very effective controls for infrared reflectance and transparency, and they could control heat reflectance, resulting in a significant decrease in surface temperature.

**Keywords:** infrared reflection; infrared reflective pigments and coatings; total solar reflection; infrared solar reflection; refractive index; chromatic properties; brightness; heat reflection





## 1. Introduction

In cases of surfaces exposed to sunlight, solar energy can be transmitted, reflected, or absorbed. When the frequency of the incoming light is near to the electron energy levels of a sun-exposed material, the electrons will absorb the light wave energy and the electrons' energy state changes. In other words, the absorbed light is converted into thermal energy. The absorption of light depends on the nucleus and on electrons. In its transmission, light moves through a substrate, and at the reflection of different wavelengths, the angle of incidence of the light is equal to that of the reflection on smooth surfaces; consequently, the light bounces back from the surface [1].

More solar energy absorbed requires more cooling for the interior. Solar irradiation comprises three regions: ultraviolet (280–400 nm, only 5% of the sun's energy), visible (400–700 nm, 43% of the sun's energy), and near infrared (700–2500 nm, 52% of the sun's energy). Near infrared light plays an important role in heat generation [2,3]. Absorbed sunlight increases temperature and requires more cooling energy [4,5], additionally it can cause physical damage.

Energy consumption could be diminished not only by cooling but also by using special infrared reflective coatings. When solar reflectance increases, surface temperature diminishes; solar radiation is reflected rather than absorbed. Consequently, an object

retains a more comfortable thermal condition without air conditioning [6,7]. In the case of traditional coatings, the main consideration is often its aesthetic property and not its infrared reflective character. When infrared reflective coatings are used, the most important features are their heat reduction and energy savings, as well as improving indoor comfort. Of course, aesthetic appearance is also important. The proper selection of an infrared reflective coating needs systematic screening of its refractive index, transparency, electric resistance, thermal conductivity, etc.

The use of color pigments goes back to antiquity; they formed the basic part of all paints and decoration [8–10]. Pigments should be stable and chemically inert to stand up against aggressive environmental conditions and in order to retain their color.

Generally, inorganic pigments are small particles that are insoluble in solvents and binders [11]. A lot of color pigments (e.g., white, blue, yellow) that are responsible for the color of objects absorb visible light and reflect others selectively (especially in the near infrared waves). Black pigments absorb and store heat not only in the visible but also in the near infrared wavelength, too [12].

Generally used infrared reflecting pigments applied in the paint industry are summarized in Table 1.

**Table 1.** Infrared reflective pigments used in the coatings.

| General Description | Materials in the Coating | References |
|---|---|---|
| Metals | Aluminum, aluminum foil, metallized thin films | [13–15] |
| Metal oxides | $TiO_2$, $ZnO$, $CuO$ | [16,17] |
| Inorganic pigments | $Fe_2O_3$, $Cr_2O_3$ $ZnS$, $Sb_2O_3$, $ZrO_2$, $SrCuSi_4O_{10}$, $Nd_2Si2O_7$, $BiVO_4$ $Cr_2O_3$, $BiFeO_3$, $Ce4Si_3S_{12}$ | [18] |
| Composites: oxide-metal oxide | $TiO_2/Ag/TiO_2$, $ZnS/Ag/ZnS$ | [19] |
| Single layers and stacked polar dielectrics | $SiC$, $GaN$ | [20] |
| Single and multilayers | Ag doped hafnium nitride | [21,22] |
| Small spheres | glass, glitter, crystals | [16,22–25] |

Coatings of metals (silver, aluminum, gold) are very useful because of their high reflectivity in the infrared region; they are mainly used for windows. Metal oxides ($TiO_2$, $ZnO$) act in very thin polymer films blended with other materials. Compatibility and durability are important factors in the selection of materials [26]. $TiO_2$ (rutile, anatase) is mainly applied in white pigments [27,28]. The application of $TiO_2$, together with other pigments, could enhance infrared reflective efficacy [29,30]. The usefulness and effectiveness of zinc oxide [31,32], and the importance of bismuth vanadate [33] and bismuth titanate [34,35], as well as of the rare earths [36,37], has also been demonstrated. A very useful review summarizes the optical properties of near infrared reflective inorganic pigments published in 2019 [7]. Other publications describe and sort the most important pigments (according to their color) active in the near infrared light region [38–41]. Though organic pigments are brighter and more transparent than inorganic ones, their light resistance is much less, and they can migrate in thermoplastics.

In order to compare the usefulness of infrared reflective coatings with other possibilities that can also reduce heat absorption and improve energy efficiency, we have to mention other resolutions like:

- Low-emissivity coatings: these can reflect and absorb infrared and block UV radiation. They effectively control the heat exchange between the interior and exterior of buildings, but an infrared reflective coating reduces heat absorption [42,43].

- Thermochromic coatings: These change their color and/or their opacity due to altered temperature; in this way, they control the amount of heat that enters buildings. But, as mentioned in the previous case, infrared reflective coatings reduce heat absorption more significantly [44,45].
- Phase-change materials: some materials, when they change their phase from liquid to solid, absorb or release heat, regulate temperature, and reduce energy consumption in a building. Though the final effect of these materials and infrared reflective coatings are similar, the mechanisms for how they act and where they can be used are different. The application of an infrared reflective coating is more extensive [46,47].

The selection of the best infrared reflective coating needs systematic screening. The pigments should be compatible with any other ingredients in a coating; they could be used in combination with different pigments. The particle size of pigments has an important impact on the refractive activity (they must be in the range between 0.35 and 0.55 μm).

The cost, compatibility, performance, and aesthetics, as well as environmental impact and energy efficiency, determine the applicability of infrared reflective pigments. These pigments could have the following arrangements:

- One-layer system: the coating is applied directly onto a solid substrate, which contributes to an increase in the refractivity of the coating [18,48].
- Two-layer system: this is the more often used technique; a top coat with infrared transparency contains the proper pigment and is deposited onto a primer. The sun's radiation influences first the upper layer [49].
- Multilayer structure: multiple layers with different refractive properties are deposited onto each other. The interference effect between the layers increases their reflective properties. It is important to take into consideration not only the chemical properties but also the thickness of each layer [50].

Generally in the case of paints, the lifespan of an infrared reflective coating depends on different factors like UV radiation, surface preparation, moisture, chemical resistance, etc. The proper selection of pigments and ingredients can enhance the lifespan of the coated surfaces and decrease energy consumption.

The aim of this work was to elaborate on infrared reflective as well as transparent coatings. The importance of these types of coatings has already been detailed. In our experiments, as a model, the RAL7016 Anthracite grey color was chosen, as this is a favored color used on vehicles and buildings (that are irradiated by sunshine), but this pigment does not have infrared reflective properties (as will be demonstrated in our results). In order to have climatic comfort in vehicles and buildings, it is necessary to use those types of coatings that have the same color (Anthracite grey); at the same time, they can reduce interior temperatures and lower energy consumption, i.e., they are more cost-effective. In the present work, coatings with high infrared reflection and transparency and good UV resistance were elaborated in the selected RAL7016 Anthracite grey color. In order to demonstrate that not only white but other colors, too, can reflect sunlight, six colored coatings with identical colors in the visible wave range were formulated. Pigments traditionally used in the paint industry have been replaced by others that have similar colors but have much higher infrared reflection. After selection of the proper ingredients, the pigments were mixed in water-based coatings and were developed on two primers: one of them with good (white) and the other with bad (black) reflectance. The influence of the pigments and other components in the coatings were characterized and compared in order to select the most effective infrared reflecting ones.

## 2. Materials and Methods

### 2.1. Composition of the Paints

In all these compositions, the applied pigments produced the Anthracite grey color of RAL7016 (Table 2). These pigments were either inorganic or organic, and in some cases a composition of two pigments. In all cases they were mixed with $TiO_2$ to achieve the Anthracite grey color.

**Table 2.** Compositions of coating samples under investigation.

| Name of the Coating | Short Name of Coatings | Pigment | Concentration of Pigments in Coatings | Characteristics of the Coating |
|---|---|---|---|---|
| Metallux Aqua 2k SZ RAL7016 L95/CR28 | L95/CR28 | - Inorganic metal-oxide black pigment—rutile titanium dioxide pigment | 20% | infrared reflective covering coating |
| Metallux Aqua 2k SZ RAL7016 S84/CR28 | S84/CR28 | - Perylene black pigment<br>- rutile titanium dioxide pigment | 17% | infrared transparent covering coating |
| Metallux Aqua 2k SZ RAL7016 S84/CR28Lacquer | S84/CR28/Lacquer | - Perylene black pigment<br>- rutile titanium dioxide pigment | 6% | infrared transparent coating with optimal pigment concentration |
| Metallux Aqua 2k SZ RAL7016 TR RU/ZD | TR RU/ZD | - Diketo-Pyrrolo-Pyrrole pigment (CI.264)—Cu-Phtalocyanine-pigment (IR transparent, opaque)<br>- rutile titanium dioxide pigment (IR Reflective, covering) | 20% | infrared transparent opaque coating |
| Metallux Aqua 2k SZ RAL7016 PS/ZD | PS/ZD | - Diketo-Pyrrolo-Pyrrole pigment (CI.254)<br>- Cu-Phtalocyanine-pigment<br>- rutile titanium dioxide pigment | 21% | infrared transparent opaque covering coating |
| Metallux Aqua 2k SZ RAL7016 S84/A80 | S84/A80 | - Perylene black pigment<br>- special titanium dioxide pigment | 10% | infrared transparent opaque coating |
| Metallux Aqua 2k SZ RAL7016 | Traditional | - carbon black pigment (IR absorber, covering)<br>- rutile titanium dioxide pigment | 19% | Generally used in the paint industry, not infrared reflective |

Metallux Aqua 2k: OH-acrylate-isocyanate resin, two-component coating material. RAL7016: One of the most popular shades, also known as Anthracite Grey.

The surface coatings with pigments were deposited on primers (white or black). The top coatings made of Metallux Aqua 2k (which is an OH-acrylate-isocyanate, two-component, water-soluble polyurethane system with high UV and chemical resistance and excellent mechanical properties produced at EGROKORR Ltd., Érd, Hungary) were compounded with different pigments, fillers and binders. $TiO_2$ helped to attain the proper grey color.

### 2.2. Characterization of Coatings with Infrared Transparent Pigments

Refractive index: this is an important parameter of pigments. It measures the reflectivity of coatings, either at a given wavelength or in a range of wavelengths. The reflection of all coated samples was measured in the 335 to 2500 nm wavelength range (Surface Optics SOC410). The total solar reflection (TSR) and the infrared solar reflection (IR-SR: 700–2500 nm) vales are summarized in tables and demonstrated in figures.

Characterization of chromatic properties: For the chromatic characterization of all the coatings, a RAL7016 color card was used. The brightness was measured using a Konica Minolta CM-25cG instrument.

The colors of samples were determined according to the CIE, as shown in Figure 1 (Commision Intenationale de l'Eclairage).

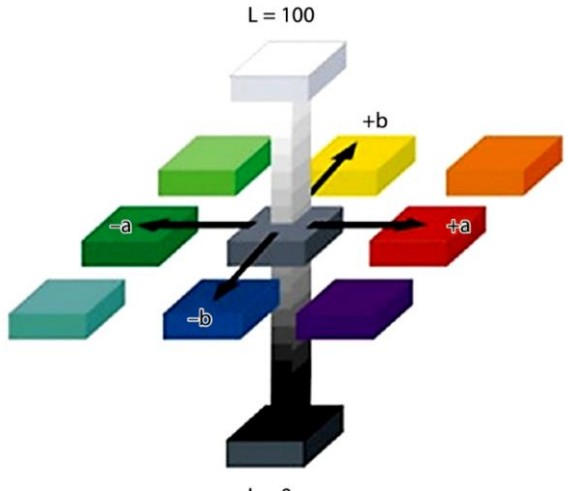

**Figure 1.** Colorimetric coordinates, CIE color system L*, a*, b*. L*: represents the differences between light (100%) and dark (0%). a*: displays the difference between red (+a*) and green (−a*). b*: corresponds to the difference between yellow (+b*) and blue (−b*).

The color differences (ΔE*) between two color points are calculated according to the distance between their locations in the three-dimensional space (Table 3).

**Table 3.** The color differences and their consequence.

| Measured ΔE* | Visual Observation |
| --- | --- |
| ΔE* < 1.5 | not observable |
| 1.5 < ΔE* < 3 | observable |
| 3 < ΔE* < 6 | well observable |
| 6 < ΔE* | significant |

Heat camera: This instrument can evaluate temperature and show the change in temperature caused by infrared reflection. It can display a thermal image over and under the coated surfaces, and it compares the measured values to the ambient temperature. The instrument is a GUIDE PS610.

## 3. Results and Discussions

### 3.1. Chromatic Parameters

The chromatic (L*, a* and b*) and brightness values are summarized in Table 4.

**Table 4.** Colorimetric and brightness values measured on coatings applied on white and black primers.

| Short Name of Coating | Substrate | ΔE* | Δa* | Δb* | ΔL* | Brightness [%] |
| --- | --- | --- | --- | --- | --- | --- |
| Traditional | white | 0.560 | 0.440 | 0.270 | −0.230 | 84,990 |
|  | black | 0.500 | 0.400 | 0.290 | −0.080 | 84,750 |
| L95/CR28 | white | 1.300 | 1.050 | 0.020 | −0.760 | 84,750 |
|  | black | 1.420 | 1.100 | 0.020 | −0.900 | 84,090 |
| S84/CR28 | white | 0.750 | −0.340 | −0.350 | −0.570 | 49,810 |
|  | black | 0.880 | −0.410 | −0.360 | −0.690 | 67,760 |

**Table 4.** *Cont.*

| Short Name of Coating | Substrate | ΔE* | Δa* | Δb* | ΔL* | Brightness [%] |
|---|---|---|---|---|---|---|
| TR RU/ZD | white | 1.580 | −1.260 | 0.110 | 0.940 | 88,190 |
| | black | 1.320 | −1.180 | 0.070 | 0.590 | 89,960 |
| S84/CR28/Lacquer | white | 1.940 | 0.000 | 0.290 | −1.920 | 89,270 |
| | black | 2.010 | −0.270 | 0.210 | −1.980 | 89,230 |
| PS/ZD | white | 2.150 | −0.180 | −1.990 | −0.810 | 89,520 |
| | black | 2.110 | −0.160 | −1.980 | −0.700 | 88,450 |
| S84/A80 | white | 2.530 | −0.410 | −0.390 | −2.470 | 89,210 |
| | black | 2.660 | −0.480 | −0.370 | −2.590 | 89,350 |

The chromatic parameters clearly show that there are no significant differences between color and brightness values, whether the coating was applied on white or black primer, i.e., in both cases these coatings could properly cover the substrate; there were no significant differences in color parameters. On the other hand, using black and white backgrounds beneath the coatings, we wanted to see which composition was the best or worst for infrared light reflection. The difference in the brightness parameters was due to the fact that in one case the coatings were applied on white, in the other case on black, primer. It is clear that the brightness depends on the primer. The white one reflects the infrared waves; the black one absorbs them. On the other hand, the difference in the brightness parameters could depend on the composition: the S84/CR28 coating shows less brightness on both primers than the S84/CR28/Lacquer coating, proving the importance of the concentrations of pigments, as they differ only in that. The parameters of Δa* and Δb* developed on black and white primers using the same coatings did not change significantly. In the ΔL* values measured on the same upper coat that were developed on very unlike primers, there are no remarkable differences (one exception is the "traditional" coating).

### 3.2. Reflection Behavior of Coatings

The behavior of coatings (primer plus enamel) in the 335 to 2500 nm spectrum is demonstrated in Table 5 and in Figures 2 and 3. By using a black background, we wanted to demonstrate the influence of a primer coated with infrared absorptive (low total solar reflectance) characteristics; on the other hand, the white primer had good infrared reflective (high total solar reflective) characteristics, i.e., in this case, the warming of the solid surface was much less than that of the black primer.

Results summarized in Table 5 allow the comparison of refractive values measured in the wavelength range between 335 nm and 2500 nm. At lower wavelengths, the differences between the values measured on the same upper coat developed on different primers are similar in most cases. But in the infrared wavelength range there are—in some cases significant—differences between the samples developed on white or black primer. The coatings with S84/CR28 and S84/CR28/Lacquer contain the same pigments but at different concentrations. Interestingly, the lower concentration resulted in better refractive values. This means that the production of the S84/CR28/Lacquer is more cost effective, and additionally shows the same infrared reflective characteristic. The S84/A80 sample can also effectively reflect infrared light; it differs from the other two in appearance: it is infrared opaque.

Analyzing the curves in Figure 2, it is clear that the traditional coating shows the worst reflective characteristics. The best results were obtained by the coatings S84/CR28 and S84/CR28/Lacquer. Similarly, a good result was measured for the S84/A80 coating. In other words, the best results were provided by coatings with infrared transparent properties.

**Table 5.** Reflection behavior of coatings applied either on white or on black primers, measured in the large wavelength range (UV: 335–380 nm; visible: 400–720 nm; near infrared: 700–2500 nm).

| Short Name of Coating | Substrate | Reflection Values Measured at Different Wavelengths [nm] | | | | | | |
|---|---|---|---|---|---|---|---|---|
| | | 335–380 | 400–540 | 480–600 | 590–720 | 700–1100 | 1000–1700 | 1700–2500 |
| | | 1 | 2 | 3 | 4 | 5 | 6 | 7 |
| Traditional | white | 0.061 | 0.085 | 0.082 | 0.075 | 0.067 | 0.062 | 0.062 |
| | black | 0.059 | 0.085 | 0.082 | 0.075 | 0.067 | 0.063 | 0.062 |
| L95/CR28 | white | 0.063 | 0.119 | 0.097 | 0.099 | 0.283 | 0.730 | 0.635 |
| | black | 0.063 | 0.115 | 0.095 | 0.097 | 0.258 | 0.438 | 0.324 |
| S84/CR28 | white | 0.067 | 0.2 | 0.137 | 0.175 | 0.810 | 0.813 | 0.564 |
| | black | 0.066 | 0.147 | 0.113 | 0.142 | 0.489 | 0.331 | 0.136 |
| TR RU/ZD | white | 0.058 | 0.170 | 0.122 | 0.113 | 0.544 | 0.795 | 0.589 |
| | black | 0.060 | 0.147 | 0.111 | 0.103 | 0.401 | 0.395 | 0.169 |
| S84/CR28/Lacquer | white | 0.070 | 0.192 | 0.134 | 0.180 | 0.809 | 0.795 | 0.553 |
| | black | 0.068 | 0.119 | 0.100 | 0.124 | 0.334 | 0.197 | 0.085 |
| PS/ZD | white | 0.064 | 0.167 | 0.118 | 0.104 | 0.533 | 0.790 | 0.586 |
| | black | 0.060 | 0.134 | 0.103 | 0.090 | 0.318 | 0.281 | 0.112 |
| S84/A80 | white | 0.057 | 0.184 | 0.125 | 0.135 | 0.723 | 0.788 | 0.602 |
| | black | 0.056 | 0.120 | 0.096 | 0.100 | 0.332 | 0.311 | 0.223 |

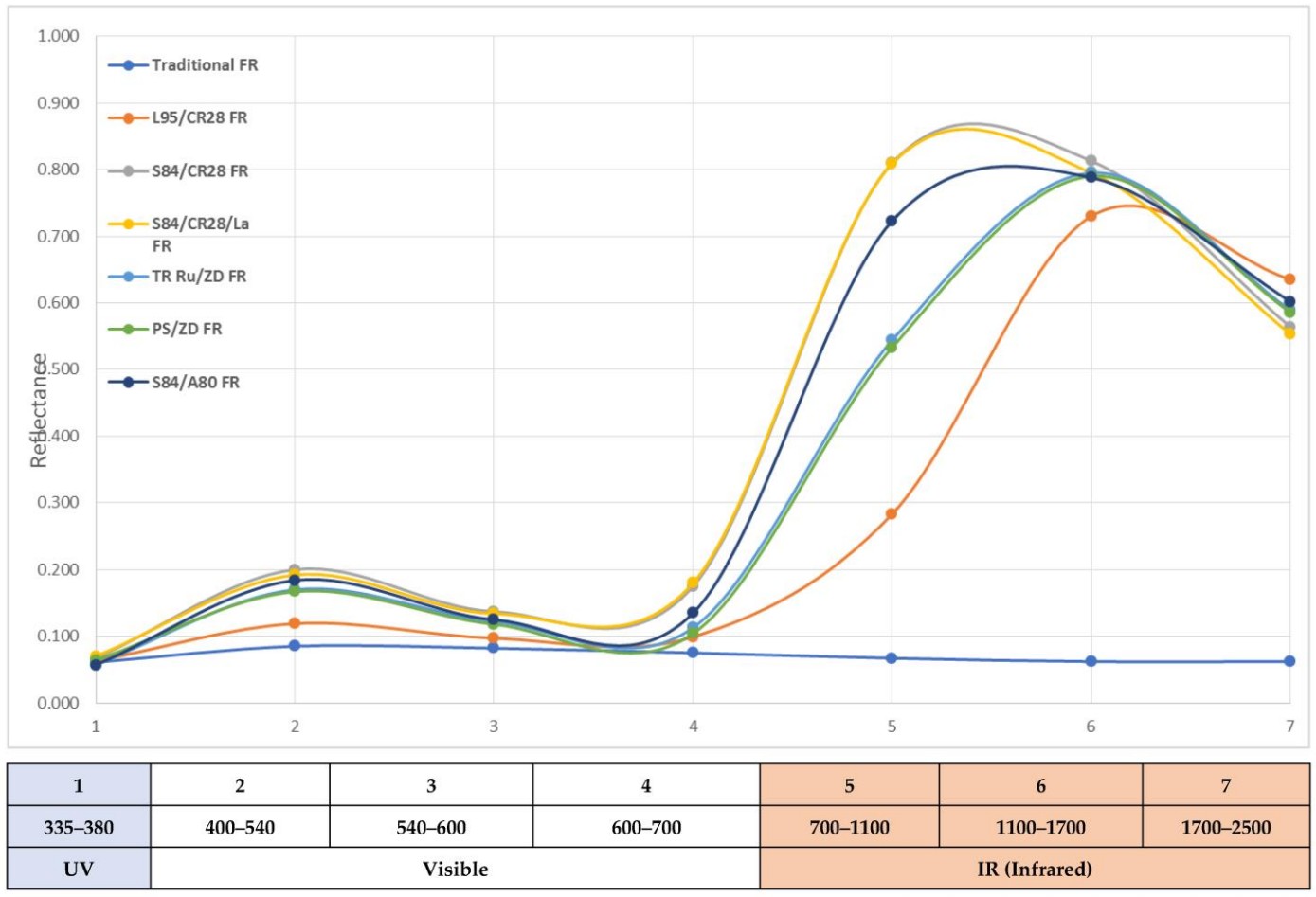

**Figure 2.** Reflection behavior of different coatings applied on white primer measured in the wavelength range between 335 nm and 2500 nm (FR: coatings developed on white primer; La: Lacquer).

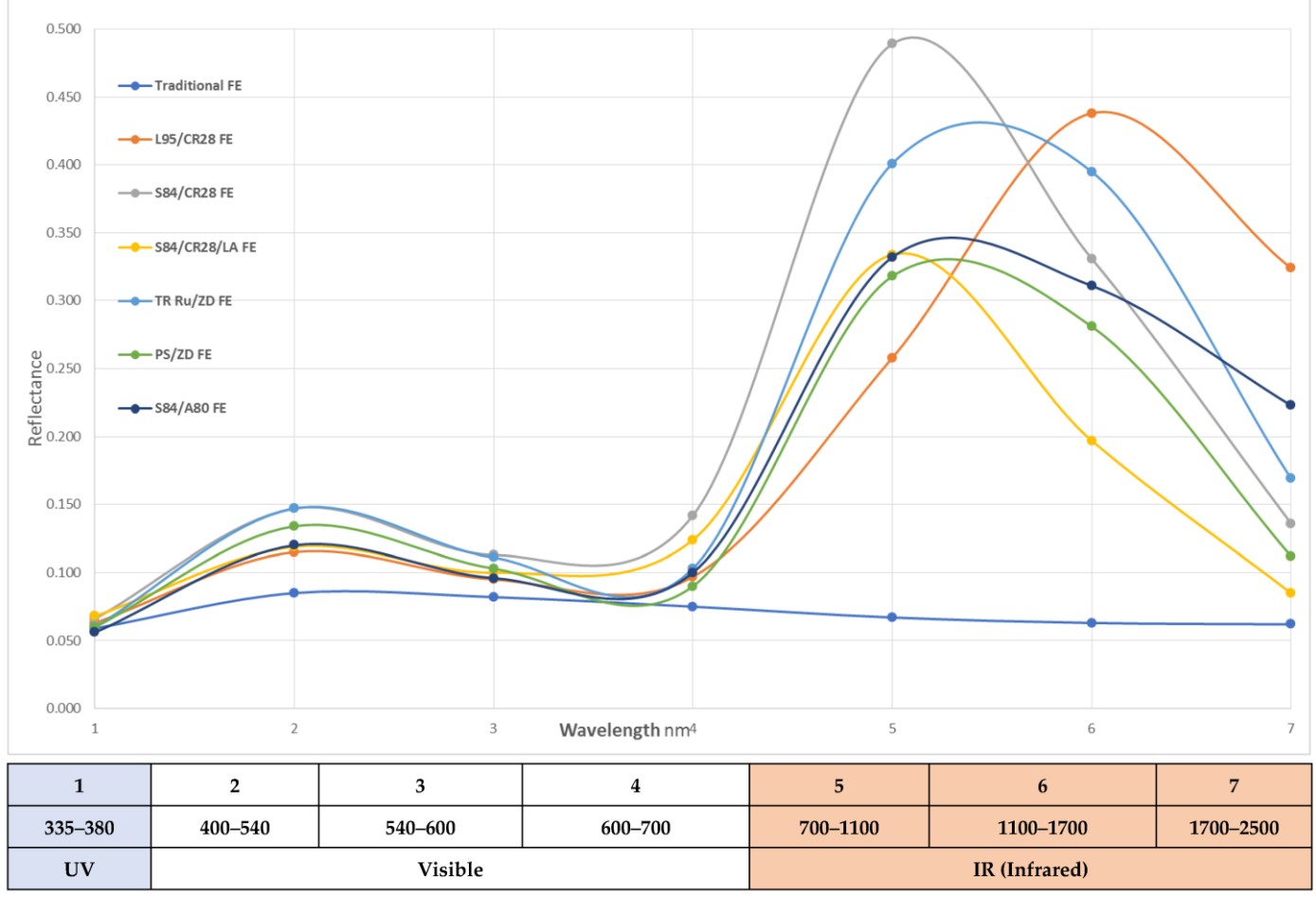

| 1 | 2 | 3 | 4 | 5 | 6 | 7 |
|---|---|---|---|---|---|---|
| 335–380 | 400–540 | 540–600 | 600–700 | 700–1100 | 1100–1700 | 1700–2500 |
| UV | Visible | | | IR (Infrared) | | |

**Figure 3.** Reflection behavior of different coatings applied on black primer measured in the wavelength range between 335 nm and 2500 nm (FE: coatings developed on black primer; La: Lacquer).

The curves in Figure 3 prove that the best results for coatings developed on a black background were measured on those that contain transparent infrared pigments. These were followed by coatings that contained top infrared pigments. The worst result was measured on the coating with traditional pigment.

On the basis of the results measured on white and black backgrounds (Figures 2 and 3), it is clear that coatings on white primer always show higher infrared reflectance in the infrared range than on black primer. The only exception is in coatings with traditional pigments, where there is no difference caused by the different color of the primers. In order to get a coating with desired total infrared reflective ability, the use of a white primer with a high total reflectance pigment is advisable.

### 3.3. Total Solar Reflectance (TSR) and Infrared Solar Reflectance (IR-SR) Parameters

Measuring the parameters of TSR and IR-SR allowed for a numerical evaluation of these important factors and helped to obtain a more clear comparison of the samples under investigation.

Spectral solar irradiation $i_{solar}$ plays important role in the calculation of solar reflectance; $\rho_{solar}$ (solar absorbance $\alpha_{solar}$) is calculated from the measured spectral reflectance $\rho(\lambda)$:

$$\rho_{solar} = \frac{\int_{\lambda_1}^{\lambda_2} \rho(\lambda) \times i_{solar} \, d\lambda}{\int_{\lambda_1}^{\lambda_2} i_{solar} \, d\lambda} \tag{1}$$

Initialization: G173GT (see Figure 4); source: ASTM Standard G173-03 (2012) [51]; Clear sky AM1.5 global irradiance on sun-facing surface tilted at 37°.

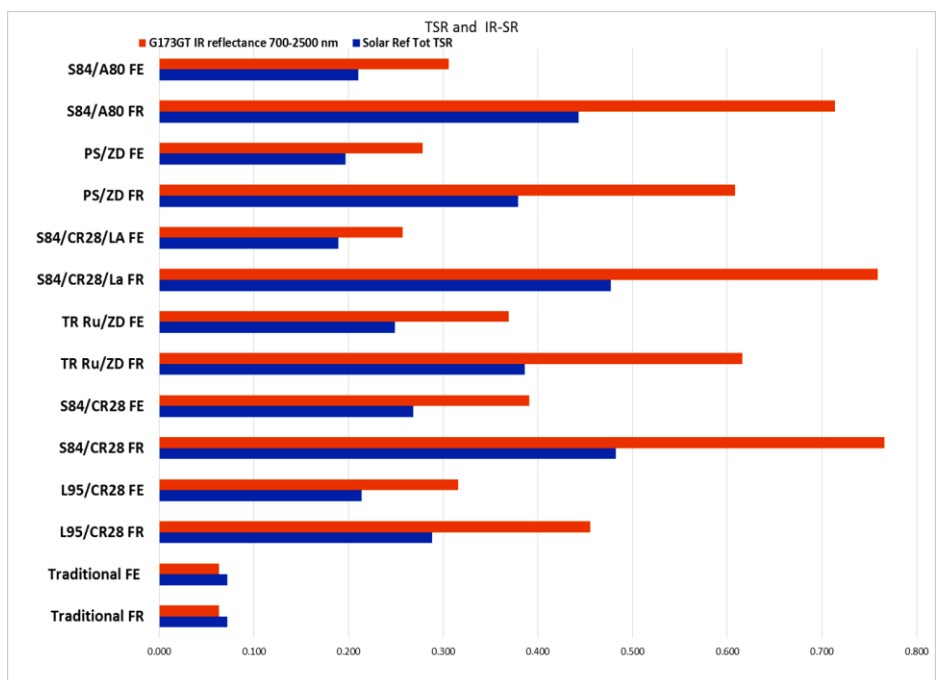

**Figure 4.** Summary of the total solar reflectance and the infrared solar reflectance values measured on the coatings developed according to the ASTM Standard G173-03 (FE: coatings developed on black primer; FR: coatings developed on white primer; La: Lacquer).

Numerical values for the total solar reflectance (TSR) and the infrared solar reflectance (IR-SR) depicted in the same diagram (Figure 4) allow for a better comparison of the heat-reflective usefulness of the coatings. In the case of the coatings developed with the RAL7016 Anthracite grey color, there are significant differences between the infrared solar reflectance and the total solar reflectance.

According to Figure 4, the worst results were obtained with a coating using traditional pigments. Coatings that contained covering infrared reflective pigments produced better results. The best examples are those that were made of infrared transparent pigments, represented by the L95/CR28, S84/A80, S84/CR28, and S84/CR28/Lacquer compositions. The columns in Figure 4 prove that the white primer is beneficial when a coating with high total solar reflection is required. In all cases, the infrared solar reflection values are higher than the total solar reflection ones. This means that the coatings under investigation in the wavelength range between 700 and 2500 nm show better reflection than in the full spectrum (330–2500 nm), which contains the absorption of UV and visible light.

Table 6 demonstrates more clearly the differences between the TSR and the ISR values by numerical value.

As all coatings used Anthracite grey (RAL7016), they were unable show differences in the visible wavelength, only in the infrared range. A high reflection value (close to 100) means significant light reflection, which means that the surface does not really absorb infrared light, and that is why it is much less warm. We could reach this aim using coatings S84/CR28 (TSR: 48.2%; IR-SR: 76.6%) and of S84/CR28/Lacquer on white primer (TSR: 47.7%; IR-SR: 75.9%); on the other hand, these coatings on black primer are less effective. The worst results were observed using traditional coatings (TSR: 7.3%, IR-SR: 6.3%).

**Table 6.** The TSR and IR-SR values measured on different coatings.

| Short Name of Coating | Primer | TSR % | IR-SR % |
|---|---|---|---|
| traditional | white | 7.2 | 6.3 |
| | black | 7.2 | 6.3 |
| L95/CR28 | white | 28.8 | 45.5 |
| | black | 21.4 | 31.6 |
| S84/CR28 | **white** | **48.2** | **76.6** |
| | black | 26.8 | 39.1 |
| TR RU/ZD | white | 38.6 | 61.6 |
| | black | 24.9 | 36.9 |
| S84/CR28/Lacquer | **white** | **47.7** | **75.9** |
| | black | 18.9 | 25.7 |
| PS/ZD | white | 37.9 | 60.8 |
| | black | 19.7 | 27.8 |
| S84/A80 | white | 44.3 | 71.4 |
| | black | 21.0 | 30.6 |

### 3.4. Evaluation of Heat Reflection

In order to demonstrate the influence of a coating on the substrate's temperature, samples were placed on racks outside and exposed to sun. The coated panels' temperatures were measured via heat camera (Table 7), and the temperature values were analyzed using the GUIDE PS610 instrument.

**Table 7.** Temperature values measured on coatings via heat camera.

| Name of the Coating | Primer | Temperature [°C] | | |
|---|---|---|---|---|
| | | Max | Min | Average |
| Traditional | white | 64.5 | 61.7 | **60.0** |
| | black | 64.2 | 52.4 | **60.4** |
| L95/CR28 | white | 59.3 | 48.6 | 54.5 |
| | black | 61.5 | 51.6 | 57.1 |
| S84/CR28 | white | 52.2 | 44.3 | **48.5** |
| | black | 59.6 | 50.9 | 55.5 |
| TR RU/ZD | white | 57.4 | 48.0 | 53.2 |
| | black | 61.4 | 51.7 | 57.3 |
| S84/CR28/Lacquer | white | 54.2 | 45.2 | **49.8** |
| | black | 61.7 | 52.4 | 58.3 |
| PS/ZD | white | 55.5 | 47.2 | 53.0 |
| | black | 61.0 | 51.7 | 57.6 |
| S84/A80 | white | 52.3 | 46.2 | **49.8** |
| | black | 59.4 | 51.4 | 56.0 |

The values described in Table 7 allow for a comparison of the effectiveness of coatings from the point of view of heat reflection efficiency. As the red-marked numbers show, coatings **S84/CR28, S84/CR/Lacquer**, and **S84/A80,** developed on white primer, showed the best heat reflective characteristics. Comparing the temperature values with those measured on traditional coating, the highest difference we achieved was **11.5 °C**, which is a significant value. The temperature differences measured on the good samples were in the range of the accuracy of the device; in other words, their effectiveness as infrared reflective coatings is similar.

The results obtained via different techniques (colorimetric, total solar reflectance, and infrared solar reflectance parameters; heat reflection values) showed that the best infrared

reflective coatings were developed on white primer, as they always showed higher infrared reflectance than those covered in black primer. The results obtained via the other different methods support this observation. Coatings with special pigments have better reflection in the wavelength range between 700 and 2500 nm than in the full spectrum (330–2500 nm), which contains the absorption of UV and visible light, too. Considering the composition of the best infrared reflective coating, it proved the importance of pigment concentration: a decrease in the concentration resulted in much better characteristics in the case of the S84/CR28 and S84/CR28/Lacquer coatings. This factor is important from the point of view of coating production: les pigments make the product more cost effective. We can draw another inference provided by these results: the infrared transparent coatings ranked first among the considered coatings.

We anticipate that these coatings with high infrared reflective compositions will be extensively applied in the future given their significant energy and costs savings compared with currently used materials.

## 4. Conclusions

The paper discusses the elaboration of different coatings with infrared reflectivity in order to reduce surface temperature, decrease energy consumption, and increase material lifespan. All the ingredients were mixed into a two-component, acrylate–isocyanate resin. RAL7016 Anthracite grey was selected as a color as it is often applied to vehicles and buildings, but it is not infrared reflective. In our coatings, the pigments that produced the same color were either organic or inorganic in origin. Six different coatings (either infrared reflective or infrared transparent) were developed and investigated, either on white or on black primers, and were characterized using colorimetric and brightness values, refractive parameters (in the UV, visible, and infrared wavelength range, focusing on total solar reflectance and infrared solar reflectance), and by measuring the decrease in the surface temperature generated by the coatings and recorded via a heat camera. In all cases, a traditional, but not infrared, reflective coating was the control.

The experimental results proved: The chromatic parameters did not depend significantly on the primer. Two of the coatings (S84/CR28/Lacquer and S84/A80) showed high brightness. Their pigments differ in the $TiO_2$ crystal form.

Significant differences in the refractive values measured between the 335 nm and 2500 nm wavelength range are only in the infrared range. In some cases, there are differences caused by the primer. Very good infrared reflecting properties were measured on samples S84/CR28 and S84/CR28/Lacquer; their composition was similar, different only in pigment concentration. The S84/A80 coating produced similarly good infrared reflecting results, but this coating was opaque.

Analyzing the total solar reflectance and the infrared solar reflectance parameters measured on all coating samples, high values were achieved by samples S84/CR28 (TSR: 48.2%; IR-SR: 76.6%) and S84/CR28/Lacquer (TSR: 47.7%; IR-SR: 75.9%) on white primer; on the other hand, these coatings were less effective on black primer. The worst results were produced by the traditional coating (TSR: 7.3%, IR-SR: 6.3%). Though the S84/A80 coating showed similarly good results compare to the S84/CR28/Lacquer, it contained a special infrared transparent binder, and its production process is much more difficult.

High heat reflecting properties were measured on those coatings that gave the best results in the previously mentioned tests: S84/CR28, S84/CR28/Lacquer and S84/A80. A high temperature decrease of 11 °C was achieved on the coated surfaces, a remarkable savings in energy consumption.

**Author Contributions:** Conceptualization, J.M.; Validation, A.-E.B.; Formal analysis, L.T.; Writing—original draft, J.T. All authors have read and agreed to the published version of the manuscript.

**Funding:** This research had no funding.

**Institutional Review Board Statement:** Not applicable.

**Informed Consent Statement:** Written informed consent has been obtained from the patients to publish the paper.

**Data Availability Statement:** The data presented in this study are available on request from the corresponding author.

**Conflicts of Interest:** The authors declare no conflict of interest. The funding sponsors had no role in the design of the study; in the collection, analyses, or interpretation of data; in the writing of the manuscript, and in the decision to publish the results.

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
