# Peer review of "Development of Effective Infrared Reflective Coatings"

_applsci, doi:10.3390/app132312903_

Round 1

Reviewer 1 Report

Comments and Suggestions for Authors

This paper investigated the relationship between infrared reflection, surface temperature rise and the concentration and composition of pigments in coatings. By investigating different coating compositions, the pigments and the binding systems were optimized. But there are still some drawbacks. In view of the following questions (not just the given examples), I still suggest it should be in minor revision again before its publish in this journal.

Question 1: There are some format errors, such as he format of the author's third unit is not uniform; In Abstract, the font size of the first and second paragraphs is different.

Question 2: In Introduction, when the surface is exposed to sunlight, solar energy can be transmitted, reflected and absorbed. The mechanism should be further explained, and references should be added.

Question 4: There are too many paragraphs in the Introduction and the reading feels disconnected.

Question 5: Page 2, L 28-40, when listing the commonly used infrared reflective coatings, it is more concise and clear to use a table.

Question 6: Page 3, L 21-28, when talking about the factors that affect infrared reflectivity, references should be added.

Question 7: Page 3, L 32-41, references should be added when referring to the arrangement of infrared reflective pigments.

Question 8: Page 4, in the Materials and Methods section, when listing the coating samples under investigation, using tables is more intuitive.

Question 9: Page 4, in the Characterization of chromatic properties, the letters in the diagram are not clear enough, it is better not to overlap with the diagram, and should be annotated, such as “Figure 1”.

Question 10: Page 5, the first table is not annotated.

Question 11: Page 5, Table 1, there is a big difference in the brightness value of the coating on the black and white substrate, which is not consistent with the expression in the paper.

Question 12: Page 5 and page 6, the placement of Tables 1 and 2 is incongruous.

Question 13: Page 6, Figure 1 is too vague and lacks clarity.

Question 14: Page 7, L 1, “Figure 1. clearly demonstrates that the traditional coating with traditionally used pigment” can be deleted.

Question 15: Page 7, Figure 2 is too fuzzy.

Question 16: Page 8, the two groups of coatings with good light reflection in Table 2 can be highlighted for easier detection.

Question 17: Page 9, Table 4, temperatures measured by heat camera on the coatings were under investigation, but the maximum and minimum temperature values were reversed.

Question 18: In Conclusions, the summary is too scattered, lacks coherence, and the ending has no vision.

Question 19: In Reference citations, some research papers are too old and can use the latest reports.

Comments on the Quality of English Language

The language should be carefully improved.

Author Response

Response to Reviewer 1 Comments
1. Summary
Thank you very much for taking the time to review this manuscript. Please find the detailed
responses below and the corresponding revisions/corrections highlighted in the re-submitted
files
.
2. Questions for General
Evaluation
Reviewer’s Evaluation Response and Revisions
Does the introduction provide
sufficient background and include
all relevant references?
Must be improved The Introduction was
improved.
Are all the cited references relevant
to the research?
Can be improved The revision of the
References was done.
Is the research design appropriate? Can be improved
Are the methods adequately
described?
Can be improved The description of
methods is improved.
Are the results clearly presented? Must be improved The results are more
clearly described.
Are the conclusions supported by
the results?
Can be improved The Conclusion is
revised.
3. Point-by-point response to Comments and Suggestions for Authors
Question 1: There are some format errors, such as the format of the author's third unit is not
uniform;
In Abstract, the font size of the first and second paragraphs is different.
Response 1: The name of the second author is correct: Attila-Ede Bodnár. Thank you for pointing
this out.
The font size was unified.
Comments 2: In Introduction, when the surface is exposed to sunlight, solar energy can be
transmitted, reflected and absorbed. The mechanism should be further explained, and references
should be added.
Response 2: We have modified the text by adding explanation and reference on the three
different types of behavior of the light
. All these are marked by red color in the text.
Question 3: This question is missing in the reviewer’s list.
Question 4: There are too many paragraphs in the Introduction and the reading feels

disconnected.
Response 4: The Introduction was modified according to the Reviewer’ suggestion.
Question 5: Page 2, L 28-40, when listing the commonly used infrared reflective coatings, it is
more concise and clear to use a table.
Response 5: We accepted the Reviewer’s opinion and converted the text into a table and
completed it with some references.
Question 6: Page 3, L 21-28, when talking about the factors that affect infrared reflectivity,
references should be added.
Response 6: The authors added references about the factors that affect the infrared reflectivity.
Question 7: Page 3, L 32-41, references should be added when referring to the arrangement of
infrared reflective pigments.
Response 7: According to the Reviewer’ request literatures are inserted.
Question 8: Page 4, in the Materials and Methods section, when listing the coating samples under
investigation, using tables is more intuitive.
Response 8: The Reviewer’ suggestion was accepted and the samples under investigation are
involved in a table.
Question 9: Page 4, in the Characterization of chromatic properties, the letters in the diagram are
not clear enough, it is better not to overlap with the diagram, and should be annotated, such as
“Figure 1”.
Response 9: The authors accepted the Reviewer’s suggestion: the figure is converted into a more
visible form, and it is named “Figure”.
Question 10: Page 5, the first table is not annotated.
Response 10: The authors inserted annotation in the text.
Question 11: Page 5, Table 1, there is a big difference in the brightness value of the coating on the
black and white substrate, which is not consistent with the expression in the paper.
Response 11: The differences are due to the fact that in one cases the coating are on white, in the
other cases on black primer. It is clear that the brightness depends on the primer. The white one
reflects the infrared waves, the black one absorbs. On the other hand the differences in the
brightness parameters could depend on the composition: the S84/CR28 coating shows less

brightness on both primers then the S84/CR28/Lacquer coating demonstrating the importance of
the concentrations of components as, concerning these two compositions, they differ only in the
components concentration. The improved text marked with red color..
Question 12: Page 5 and page 6, the placement of Tables 1 and 2 is incongruous.
Response 12: The authors placed the Table 1 and 2 (in the corrected version: Table 4 and 5) onto
correct places.
Question 13: Page 6, Figure 1 is too vague and lacks clarity.
Response 13: Figure 1 (now: Figure 2) was completed in order to be more understandable.
Question 14: Page 7, L 1, “Figure 1. clearly demonstrates that the traditional coating with
traditionally used pigment” can be deleted.
Response 14: The sentence was deleted.
Question 15: Page 7, Figure 2 is too fuzzy.
Response 15: Figure 2 (now: Figure 3) was completed in order to be more understandable.
Question 16: Page 8, the two groups of coatings with good light reflection in Table 2 can be
highlighted for easier detection.
Response 16: The authors highlighted the two best infrared reflective coatings that were
analyzed.
Question 17: Page 9, Table 4, temperatures measured by heat camera on the coatings were under
investigation, but the maximum and minimum temperature values were reversed.
Response 17: The authors are very sorry for this inattention. The maximum and minimum values
are now indicated in the table correctly.
Question 18: In Conclusions, the summary is too scattered, lacks coherence, and the ending has
no vision.
Response 18: The authors revised the Conclusion in order to make it coherent, and added some
vision of the topic.
Question 19: In Reference citations, some research papers are too old and can use the
latest reports.
Response 19: Accepting the Reviewer’s suggestion the authors revised the Reference list and
completed it with additional references.

4. Response to Comments on the Quality of English Language
The language should be carefully improved
The language was surveyed by an expert in the English language.
The authors say thanks to the Reviewer for remarks and suggestions to improve the paper.

Reviewer 2 Report

Comments and Suggestions for Authors

In this paper, the influence of concentration and composition of pigments in the coatings on the infrared reflection and surface warm up was investigated. The study is relatively complete. However, there are still some problems remaining to be solved.

(1) The title of the paper needs to be revised to emphasize its main work and innovation.

(2) The abstract needs to be rewritten to highlight the main work and innovation of this article. Moreover, quantitative data can be added to verify the work of this article.

(3) The font size of the two paragraphs in abstract is inconsistent.

(4) There is formatting confusion in the main text of Page 3.

(5) The authors are suggested to replace the tables with a three-line table.

(6) Please supplement the title of the figure in Page 4.

(7) Please supplement the title of the table in Page 5.

(8) The text in Figures 1 and 2 is not clear enough, please modify.

Comments on the Quality of English Language

(1) “Results and Discussion” should be replaced by “Results and Discussions”.

(2) The title of Figure 3 can be replaced with “Summary of the total solar reflectance and the infrared solar reflectance values measured on the developed coatings”

Author Response

 1

Response to Reviewer 2 Comments
1. Summary
Thank you very much for taking the time to review this manuscript. Please find the detailed
responses below and the corresponding revisions/corrections highlighted in red color in the
re-submitted paper.
2. Questions for General
Evaluation
Reviewer’s Evaluation Response and Revisions
Does the introduction provide
sufficient background and include
all relevant references?
Can be improved The authors improved
the introduction.
Are all the cited references relevant
to the research?
Can be improved The references were
completed.
Is the research design appropriate? Can be improved It was improved.
Are the methods adequately
described?
Can be improved It was improved.
Are the results clearly presented? Must be improved The results are now
clearly presented.
Are the conclusions supported by
the results?
Can be improved The conclusion was
improved.
3. Point-by-point response to Comments and Suggestions for Authors
Comments 1: The title of the paper needs to be revised to emphasize its main work and
innovation.
Response 1: We agree with this comment. Therefore, the title was altered to a new one:
Development of effective infrared reflective coatings
Comments 2: The abstract needs to be rewritten to highlight the main work and innovation of this
article. Moreover, quantitative data can be added to verify the work of this article.
Response 2: We have, accordingly, revised the abstract according to the Reviewer’s suggestion,
and add quantitative data to evaluate spectral solar irradiance functions. It is involved in the
revised version of the paper
.
Comments 3: The font size of the two paragraphs in abstract is inconsistent.
Response 3. The authors corrected the font sizes.
Comments 4: There is formatting confusion in the main text of Page 3.
Response 4. The authors corrected the formatting that caused confusion on Page 3.

2

Comments 5: The authors are suggested to replace the tables with a three-line table.
Response 5: The authors say thank to the Reviewer for this suggestion and converted all tables
into the recommended form.
Comments 6: Please supplement the title of the figure in Page 4.
Response 6: The completion of the figure was done.
Comments 7: Please supplement the title of the table in Page 5.
Response 7: The title of the table was supplemented.
Comments 8: The text in Figures 1 and 2 is not clear enough, please modify.
Response 8: The authors accepted the Reviewer’s suggestion and modified the text in Figure 1
and 2.
4. Response to Comments on the Quality of English Language
Comments on the Quality of English Language
(1) “Results and Discussion” should be replaced by “Results and Discussions”.
(2) The title of Figure 3 can be replaced with “Summary of the total solar reflectance and the
infrared solar reflectance values measured on the developed coatings”
The authors think that the “Discussion” is generally used in singular form.
The authors say thanks to the Reviewer for the suggestion to change the title of the Figure 3 and
replaced it by the suggested text.

We would like to express our thanks to the Reviewer for the thorough survey of the paper and for
the suggestion to improve it.

Reviewer 3 Report

Comments and Suggestions for Authors

In this study, the authors fabricated various coatings for infrared light reflection. The reflective performance was measured using various techniques. The best coating could achieve a temperature reduction of 11 °C. The topic is interesting, which could attract wide readership from researchers working in this area. However, there are several issues to be addressed before its publication.

1. The introduction part is too lengthy. The authors should briefly introduce the background related to the novelty of this work.

2. The results and discussion part mainly list the results. The discussion about the results needs to be improved. In particular, the mechanism study about the high performance is lacking.

3. What do CR28 represent? It is not indicated in the text.

4. The texts in Fig. 1 and Fig. 2 are too small.

5. Materials characterization of the coatings, at least the coating with the best performance, is lacking.

6. The reason why these coatings are used for investigation needs to be described in more detail.

7. The authors are recommended to cite relevant literatures such as J. Mater. Chem. A, 2013, 1, 6416 and Phys. Chem. Chem. Phys., 2013, 15, 15499.

Comments on the Quality of English Language

N. A.

Author Response

Response to Reviewer 3 Comments
1. Summary
Thank you very much for taking the time to review this manuscript. Please find the detailed
responses below and the corresponding revisions/corrections highlighted the changes in the re
submitted files by red color
.
2. Questions for General
Evaluation
Reviewer’s Evaluation Response and Revisions
Does the introduction provide
sufficient background and include
all relevant references?
Can be improved The authors improved
the Introduction.
Are all the cited references relevant
to the research?
Can be improved The References are
revised and completed.
Is the research design appropriate? Can be improved The research design is
improved.
Are the methods adequately
described?
Must be improved The methods are more
precisely described.
Are the results clearly presented? Must be improved The discussion of results
are completed and
clearly presented.
Are the conclusions supported by
the results?
Can be improved The conclusion is
improved.
3. Point-by-point response to Comments and Suggestions for Authors
Comments 1: The introduction part is too lengthy. The authors should briefly introduce the
background related to the novelty of this work
Response 1: Thank you for pointing this out. We have shortened the Introduction and improved
the background related to the novelty. This is highlighted in the text by red color.
Comments 2: The results and discussion part mainly list the results. The discussion about the
results needs to be improved. In particular, the mechanism study about the high performance is
lacking.
Response 2: We have, accordingly, involved longer discussion of result into this section and some
explanation about the mechanisms (marked by red in the text).
Comments 3: What do CR28 represent? It is not indicated in the text.
Response 3: The CR28 is an abbreviation for composition of the sample. This is clarified in the
description of composition of coatings under investigation in Table 2.
Comments 4: The texts in Fig. 1 and Fig. 2 are too small.
Response 4: The authors completed the text.

Comments 5: Materials characterization of the coatings, at least the coating with the best
performance, is lacking.
Response 5: Thank you for pointing this out In the revised version the basic components of the
coatings are summarized and the components characterized in the Table 2. In Table 5 and 6 the
coatings with the best performance are highlighted by red color.
Comments 6: The reason why these coatings are used for investigation needs to be described in
more detail.
Response 6: This point is emphasized in the revised version, marked by red color.
Comments 7: The authors are recommended to cite relevant literatures such as J. Mater. Chem. A,
2013, 1, 6416 and Phys. Chem. Chem. Phys., 2013, 15, 15499.
Response 7: Thank you for the suggestion. The proposed papers are about the preparation of
ZnO say: “
ZnO flower hierarchical nanostructure that possess simultaneous non-light induced
superhydrophilic, antifogging and antibacterial properties, thus providing great potential in applications
such as biomedical devices, hospital building paints, and daily life uses. This demonstrated method could be
extended to fabricate hierarchical nanostructures with other microscale nanostructured materials on
various substrates for broad applications
.” In spite of the fact that both paper deal with the synthesis
of ZnO nanostructures, the authors inserted into the reference list only one of the recommended
papers that imply in the text “broad application possibility” of ZnO (e.g. it could mean their use
in infrared reflection coatings): Weiyong Yuan, Zhisong Lu, Chang Ming Li, Self-assembling
microsized materials to fabricate multifunctional hierarchical nanostructures on macroscale
substrates; J. Mater. Chem. A, 2013
1 6416–6424 *
4. Response to Comments on the Quality of English Language
Comments on the Quality of English Language
N. A.
Response: The English language is improved.
The authors say thanks for taking the time to review this manuscript in order to improve the
quality of our paper.

Reviewer 4 Report

Comments and Suggestions for Authors

The paper is interesting, but many points must be improved and addressed.

This paper reports the optimal infrared reflective coatings with different compositions by characterizing chromatic parameters, total solar reflectance, and infrared solar reflectance.

1.     The text of the introduction is too long and should be shortened. And please check the grammatical and syntax error.

2.     In Materials and Methods part, the manufacturer of Metallux Aqua 2k should be identified.

3.     To ensure reproducibility and repeatability of the experiment, the exact composition of each coating sample must be given.

4.     The author presented an abbreviation for each sample, but the terminology in manuscript, figures and tables lacks uniformity. For example, S84/CR28/Lacquer is written as S84/CR28/LAKK, S84/CR28/la, S84/CR28/lacquer, etc.

5.     An explanation of FR and FE should be provided in the captions of Figures 1 and 2, respectively. And The author divided the wavelength from 335 nm to 2500 nm into sections 1 to 7 and used this as the X-axis in Figures 1 and 2. Therefore, the x-axis caption should be modified and there should be no units.

6.     The author measured and compared color parameters, total solar reflectance, and infrared solar reflectance for each sample, but there is very little discussion of these results.

Comments on the Quality of English Language

We can understand what the author is trying to tell the reader, but it needs proper English editing.

Author Response

Response to Reviewer 4 Comments
1. Summary
Thank you very much for taking the time to review this manuscript. Please find the detailed
responses below and the corresponding revisions/corrections are highlighted in red color in the
re-submitted file.
2. Questions for General
Evaluation
Reviewer’s Evaluation Response and Revisions
Does the introduction provide
sufficient background and include
all relevant references?
Can be improved The reference list was
improved.
Are all the cited references relevant
to the research?
Can be improved The references were
surveyed and corrected.
Is the research design appropriate? Must be improved The research design was
improved.
Are the methods adequately
described?
Must be improved The description of the
methods was improved.
Are the results clearly presented? Must be improved The results are clearly
presented in the re
submitted version.
Are the conclusions supported by
the results?
Must be improved The conclusion is
reformulated and based
on the results.
3. Point-by-point response to Comments and Suggestions for Authors
Comments 1: The text of the introduction is too long and should be shortened. And please check
the grammatical and syntax error.
Response 1: Thank you for pointing this out. We agree with this comment. Therefore, we have
shortened the Introduction and checked the grammatical and syntax error.
Comments 2: In Materials and Methods part, the manufacturer of Metallux Aqua 2k should be
identified.
Response 2: We have completed the text with the name of the manufacturer of the Metallux
Aqua 2k.
Comments 3: To ensure reproducibility and repeatability of the experiment, the exact
composition of each coating sample must be given.
Response 3: We thank for this suggestion and inserted the compositions of the samples in the
Table 2.
Comments 4: The author presented an abbreviation for each sample, but the terminology in
manuscript, figures and tables lacks uniformity. For example, S84/CR28/Lacquer is written as

S84/CR28/LAKK, S84/CR28/la, S84/CR28/lacquer, etc.
Response 4: Thank you for this suggestion; we uniformed the abbreviations of the samples in the
full paper.
Comments 5: An explanation of FR and FE should be provided in the captions of Figures 1 and 2,
respectively. And The author divided the wavelength from 335 nm to 2500 nm into sections 1 to 7
and used this as the X-axis in Figures 1 and 2. Therefore, the x-axis caption should be modified
and there should be no units.
Response 5: The authors say thanks for this remark and in the re-submitted paper they have
given explanation for the abbreviations of FR and FE as well as completed the Figures with the
necessary information on the sections.
Comments 6: The author measured and compared color parameters, total solar reflectance, and
infrared solar reflectance for each sample, but there is very little discussion of these results.
Response 6: On the initiation of the Reviewer the authors discussed more detailed the results
measured on samples by different techniques.
4. Response to Comments on the Quality of English Language
Point 1:

The full text was improved by specialist in English language.

Round 2

Reviewer 2 Report

Comments and Suggestions for Authors

The authors have replied and conducted corrections according to the review comments.

Author Response

The authors say thanks for the suggestion to improve the manuscript.  In the revised version all changes are marked with red color.

Reviewer 3 Report

Comments and Suggestions for Authors

The authors have addressed most of the issues and met the publication standard of Applied Sciences.

Author Response

The authors say thanks for the acceptance of the revised version.

Reviewer 4 Report

Comments and Suggestions for Authors

This paper is suitable for publication in its current state.

Comments on the Quality of English Language

The quality of English language has been improved.

Author Response

The authors express their thanks for the previous suggestions that helped to improve the manuscript.